

# Worsening urban ozone pollution in China from 2013 to 2017 – Part 2: The effects of emission changes and implications for multi-pollutant control

Yiming Liu[1], Tao Wang[1]

[1]Department of Civil and Environmental Engineering, The Hong Kong Polytechnic University, Hong Kong, 999077, China

*Correspondence to*: Tao Wang (cetwang@polyu.edu.hk)

**Abstract.** The Chinese government launched the Air Pollution Prevention and Control Action Plan in 2013, and various stringent measures have since been implemented, which have resulted in significant decreases in emissions and ambient concentrations of primary pollutants such as $SO_2$, $NO_x$, and particulate matter (PM). However, surface ozone ($O_3$)

concentrations have still been increasing in urban areas across the country. In a previous analysis, we examined in detail the roles of meteorological variation during 2013–2017 in the summertime surface $O_3$ trend in various regions of China. In this study, we evaluated the effect of changes in multi-pollutant emissions from anthropogenic activities on $O_3$ concentrations during the same period, by using an up-to-date regional chemical transport model (WRF-CMAQ) driven by an interannual anthropogenic emission inventory. The CMAQ model was improved with regard to heterogeneous reactions of reactive gases

on aerosol surfaces, which led to better model performance in reproducing the ambient concentrations of those gases. The model simulations showed that the maximum daily 8-hour average (MDA8) $O_3$ concentration in urban areas increased by 0.46 ppbv per year (ppbv $a^{-1}$) (p = 0.001) from 2013 to 2017. In contrast, a slight decrease in MDA8 $O_3$ concentrations by 0.17 ppbv $a^{-1}$ (p = 0.005) in rural areas was predicted, mainly attributable to the $NO_x$ emission reduction. The effects of changes in individual pollutant emissions on $O_3$ were also simulated. The reduction of $NO_x$ emission increased the $O_3$ concentration in

urban areas due to the non-linear $NO_x$-volatile organic compound (VOC) chemistry and decreasing aerosol effects; the slight increase in VOCs emissions enhanced the $O_3$ concentrations; the reduction of PM emissions increased the $O_3$ concentrations by enhancing the photolysis rates and reducing the loss of reactive gases on aerosol surfaces; and the reduction of $SO_2$ emissions resulted in a drastic decrease in sulfate concentrations, which increased the $O_3$ concentrations through aerosol effects. In contrast to the unfavorable effect of the above changes in pollutant emissions on efforts to reduce surface concentrations of

$O_3$, the reduction of CO emissions did help to decrease the $O_3$ concentrations in recent years. The dominant cause of increasing $O_3$ concentrations due to changes in anthropogenic emission varied geographically. In Beijing, $NO_x$ and PM emission reductions were the two largest causes of the $O_3$ increase; in Shanghai, the reduction of $NO_x$ and increase in VOC emissions were the two major causes; in Guangzhou, $NO_x$ reduction was the primary cause; and in Chengdu, the PM and $SO_2$ emission decreases contributed most to the $O_3$ concentration increase. Regarding the effects of decreasing concentrations of aerosols,

the drop in heterogeneous uptake of reactive gases – mainly $HO_2$ and $O_3$ – was found to be more important than the increase





in photolysis rates. The adverse effect of the reductions of $NO_x$, $SO_2$, and PM emissions on $O_3$ abatement in Beijing, Shanghai, Guangzhou, and Chengdu would have been avoided if the anthropogenic VOCs emission had been reduced by 24%, 23%, 20%, and 16%, respectively, from 2013 to 2017. Our analysis revealed that the $NO_x$ reduction in recent years has helped to contain the total $O_3$ production in China. However, to reduce $O_3$ concentrations in major urban and industrial areas, VOCs emissions control should be added to the current $NO_x$-$SO_2$-PM policy.



## 1 Introduction

China has experienced severe haze pollution due to high concentrations of particulate matter (PM) in the past decade (e.g., Guo et al., 2014; Huang et al., 2014). To alleviate this air-quality problem, the Chinese government launched the Air Pollution Prevention and Control Action Plan in 2013 and has since implemented various emission control measures (Zhang et al., 2019). Anthropogenic emissions of sulfur dioxide ($SO_2$), nitrogen oxides ($NO_x$), carbon monoxide (CO), and $PM_{2.5}$ (PM with an aerodynamic diameter less than 2.5 μm) in China were reduced by 59%, 21%, 23%, and 33% from 2013 to 2017, respectively, while the emission of volatile organic compounds (VOCs) increased slightly (Zheng et al., 2018). As a result, ambient concentrations of $SO_2$, $NO_2$, CO, $PM_{2.5}$, and $PM_{10}$ (PM with an aerodynamic diameter less than 10 μm) have declined, according to data from national environmental monitoring stations (http://www.mee.gov.cn; Fig. S1). However, surface ozone ($O_3$) concentrations in urban and surrounding areas increased from 2013 to 2017 (Lu et al., 2018). It is of critical importance to evaluate the effects of the existing control policies on atmospheric $O_3$ and refine these, if necessary, to improve overall air quality.

Ground-level $O_3$ is produced by chemical reactions involving $NO_x$, CO, and VOCs in the presence of sunlight. The key step in $O_3$ formation is the oxidation of nitric oxide (NO) by hydroperoxyl ($HO_2$) and alkylperoxyl ($RO_2$) to form $NO_2$, with subsequent photolysis of $NO_2$. It is well known that the relationship between the concentrations of $O_3$ and its precursors is non-linear and that $NO_x$ can either suppress or increase $O_3$ formation depending on its abundance relative to VOCs (and CO) (e.g., Atkinson, 2000; Wang et al., 2017b). A large body of literature has established that $O_3$ formation in urban centers is generally VOCs-limited, that is, reducing VOCs emissions leads to a decrease in $O_3$ concentrations, but reducing $NO_x$ emissions has the opposite result; in contrast, $O_3$ formation above rural areas is typically in the $NO_x$-limited or transitional regime, in which reducing $NO_x$ emissions results in decreased $O_3$ (NRC, 1991; Atkinson, 2000; Wang et al., 2017b). Any process that perturbs $HO_2$ and $RO_2$ radicals will also affect $O_3$ production (e.g., Li et al., 2018a). Therefore, elucidating the chemical drivers of $O_3$ changes requires understanding the abundance and proportions of $O_3$ precursors and the radicals involved in $O_3$ formation. Aerosols in the atmosphere can affect $O_3$ concentrations via altering the solar actinic flux, which photolyzes gases to initiate oxidation (Li et al., 2011; Xing et al., 2017) and via heterogeneous reactions of reactive gases on aerosol surfaces (Li et al., 2018a; Stadtler et al., 2018; Lou et al., 2014).

Several studies have attempted to uncover the chemical drivers of the recent $O_3$ increase in China. Using a regional chemical transport model (WRF-CMAQ), Wang et al. (2019b) derived the variation of maximum daily 8-hour average (MDA8) $O_3$ concentrations due to emission changes during 2013–2015 by subtracting the simulated changes due to meteorological variations from the total observed changes. They found that the increase in $O_3$ concentrations in 2014–2015 relative to 2013 was mainly due to the emission changes, and speculated that the decrease in $PM_{2.5}$ concentrations and the reduction of $NO_x$ emission in VOCs-limited regions could be the causes. In their study, however, the effects of emission changes during the

study period were not explicitly simulated with interannual emissions. Li et al. (2019) utilized a global model (GEOS-Chem) to simulate the MDA8 $O_3$ concentrations in 2013 and 2017 and conducted sensitivity experiments for the effects of changes in

$PM_{2.5}$ concentrations and anthropogenic emissions of $O_3$ precursors ($NO_x$ and VOCs). Their results indicated that the drastic decrease in the $PM_{2.5}$ concentrations (~40%) during the period, which reduced the uptake of $HO_2$ on aerosol surfaces, was the main reason for the $O_3$ increase in the North China Plain (NCP). Wang et al. (2019a) simulated the effect of $NO_x$ emission reduction during 2012–2016 with the WRF-CMAQ model in eastern China, which indicated increasing surface $O_3$ concentrations in urban areas due to the reduction of $NO_x$ emissions. Yu et al. (2019) applied the Kolmogorov-Zurbenko

filtering technique to the observed MDA8 $O_3$ concentrations during 2013–2017 in the Yangtze River Delta region, and concluded that the changes in $O_3$ precursor emissions contributed 76.7% to the $O_3$ increase, compared with 22% due to the decrease in $PM_{2.5}$ concentration.

We have been further investigating the meteorological and chemical driver(s) of the increasing summer surface $O_3$ in urban areas of China during 2013–2017 using an improved regional chemical transport model (WRF-CMAQ) driven by interannual

meteorological data and anthropogenic emission inventories. The role of meteorological variation and total emission changes, and the effect of individual meteorological factors, are discussed in a companion paper, Part 1 (Liu and Wang, 2020). The goal of the present work is to quantify (1) the effect of the changes in anthropogenic emissions of individual pollutants ($NO_x$, VOCs, CO, PM, $SO_2$, and $NH_3$) on urban $O_3$, which has not been addressed in the aforementioned studies but is important for further development of mitigation policy and (2) the effects of changes in aerosol concentrations on $O_3$ using a regional model with

up-to-date radical sources and heterogeneous reactions. The improved model should give a more realistic account of gas-particle interactions crucial to $O_3$ formation, compared with its earlier version. In Section 2, we briefly introduce the model system and experiment setting; Section 3 first compares the simulated reactive gases that are subject to significant heterogeneous reactions with the observations reported in the literature. We then quantify the simulated trends of MDA8 $O_3$ in urban and rural areas during 2013–2017. We further investigate the response of MDA8 $O_3$ concentration to the changes in

individual pollutant emissions from anthropogenic activities from 2013 to 2017. We then examine the effect of aerosols on the $O_3$ changes by altering the photolysis rates and heterogeneous reactions. Lastly, we conduct numerical sensitivity experiments to calculate the magnitude of VOCs emission reductions needed to overcome the adverse effect of other pollutant reductions on the goal of $O_3$ mitigation. Section 4 gives the conclusions.

## 2 Methods

### 2.1 Model setting and emission inputs

The CMAQ model (version 5.2.1, the latest version) driven by the Weather Research and Forecasting (WRF) model and the interannual multi-resolution emission inventory for China (MEIC; http://www.meicmodel.org) was applied to conduct the

simulations in this study. The model settings and emission inputs are described in the companion paper (Liu and Wang, 2020).

The CMAQ model is an off-line chemical transport model (Byun and Schere, 2006) that does not consider the effect of pollutants on meteorology, but applies an in-line method (Binkowski et al., 2007) that uses the aerosol and $O_3$ concentrations predicted within a simulation to calculate the solar radiation and photolysis rates. As a result, the model takes into consideration the effect of aerosols on $O_3$ formation via altering the photolysis rates.

**2.2 Updating heterogeneous reactions**

The heterogeneous reactions in the original CMAQ model (version 5.2.1) includes only uptakes of $NO_2$, $NO_3$, and $N_2O_5$ on

aerosol surfaces. To faithfully reproduce the effect of aerosols on $O_3$ via scavenging gaseous pollutants, we updated the heterogeneous reaction rate of $NO_2$ and $NO_3$ on the aerosol surface and incorporated additional heterogeneous reactions of gases into the CMAQ model, namely the uptakes of $HO_2$, $O_3$, OH, and $H_2O_2$ (refer to Table S2 in the companion paper (Liu and Wang, 2020) for the detailed heterogeneous reactions in the original and updated CMAQ models). The uptake coefficients ($\gamma$) of gases are the key parameters of heterogeneous reactions, but they vary according to factors such as aerosol water content

and aerosol composition. In this study, we selected the "best guess" uptake coefficients for the gases, which have been widely used in chemical transport models in previous studies.

The uptake coefficient of $N_2O_5$ in the original CMAQ model was incorporated by Sarwar et al. (2012), based on the parameterization developed by Bertram and Thornton (2009) that considered its dependence on particle liquid water, particulate nitrate, and chloride.

The heterogeneous reaction rate of $NO_2$ in the original CMAQ model was suggested by Kurtenbach et al. (2001), based on the measurements at a relative humidity of 50% under dark conditions. Field and laboratory studies found that the rate not only depends on the relative humidity (Qin et al., 2009; Stutz et al., 2004) but also on sunlight intensity (Ndour et al., 2008; Stemmler et al., 2007). Fu et al. (2019) developed a new parameterization for the $NO_2$ heterogeneous reaction rate that considered both these factors, which has improved the simulation of the reaction product, nitrous acid (HONO). This

parameterization was adopted in the updated CMAQ model.

Several laboratory studies have shown that the measured $\gamma_{NO_3}$ ranges from $10^{-4}$ to $10^{-2}$ (Rudich et al., 1996; Exner et al., 1992; Moise et al., 2002). In the original CMAQ model, $10^{-4}$ was adopted as the value for $\gamma_{NO_3}$ (Mao et al., 2013). A higher value ($10^{-3}$) was recommended by Jacob (2000) and was subsequently widely adopted in chemical transport models to investigate the effect of heterogeneous reactions on $O_3$ concentrations (Stadtler et al., 2018; Lou et al., 2014). This value was adopted in

the updated CMAQ model.

The uptake coefficients of $HO_2$ vary widely, depending on the transition metal ions contained in aerosols (George et al., 2013; Huijnen et al., 2014). The heterogeneous reaction of $HO_2$ can produce either $H_2O_2$ or $H_2O$, depending on the particulate compounds in the aqueous phase. Li et al. (2019) conducted sensitivity experiments for the products of this reaction using the





GEOS-Chem model, finding little dependence on the reaction products when assessing the effect of aerosol on $O_3$

concentrations. Here, we let the heterogeneous reaction of $HO_2$ produce only $H_2O_2$, and adopt 0.2 for $\gamma_{HO_2}$, as recommended

by Jacob (2000).

We used the value of 0.1 for the uptake coefficient of OH, based on the laboratory studies of DeMore et al. (1997). This value

was also adopted by Zhang and Carmichael (1999) and Zhu et al. (2010) to explore heterogeneous reactions in a chemical

transport model.

Previous studies of the heterogeneous reaction of $O_3$ on dust have given a wide range of $\gamma_{O_3}$, from $10^{-10}$ to $10^{-4}$ (de Reus et al.,

2000; Usher et al., 2003; George et al., 2015). Bauer et al. (2004) suggested that $\gamma_{O_3}$ of $10^{-5}$ was most appropriate for model

simulations, and this value has been adopted in previous modeling studies (Liao et al., 2004; Liao and Seinfeld, 2005). We

used this value in our simulation.

DeMore et al. (1997) reported that the uptake coefficient of $H_2O_2$ on sulfuric acid and water surfaces ranged from $8 \times 10^{-4}$

to 0.18. de Reus et al. (2005) found that using accommodation coefficients of 0.2 and $2 \times 10^{-3}$ for $HO_2$ and $H_2O_2$,

respectively, ensured agreement between simulated and observed values, under the assumption that $H_2O_2$ was produced in the

heterogeneous reaction of $HO_2$. Thus, $2 \times 10^{-3}$ was adopted for the uptake coefficient of $H_2O_2$ in this study.

The companion paper (Part 1; (Liu and Wang, 2020)) comprised validation results of the updated CMAQ model against the

observations of $SO_2$, $NO_2$, CO, $O_3$, and $PM_{2.5}$ from national environmental monitoring stations. In this study, we used the

updated and original CMAQ models to simulate the concentrations of gases lost or produced on aerosol surfaces for the summer

of 2013, and compared the simulated results with the observations reported in the literature (Table S1).

**2.3 Experiment setting**

We applied the updated WRF-CMAQ model to conduct simulations for the summer months (June, July, and August) from

2013 to 2017 with anthropogenic emissions. The shipping emissions were kept unchanged in the 5-year simulation, due to a

lack of data for recent years. In Part 1 of our work (Liu and Wang, 2020), we showed the effect of changes in total anthropogenic

emission on $O_3$ changes by comparing the $O_3$ concentrations in 2013 simulated using anthropogenic emissions from different

years. In this study, three additional sets of modeling experiments were established. The first was designed to quantify the

responses of $O_3$ concentrations to changes in individual pollutant emissions from 2013 to 2017, with the simulation in 2013

being regarded as the baseline experiment. The anthropogenic emissions of $NO_x$, VOCs, $SO_2$, CO, $NH_3$, PM (comprising $PM_{10}$,

$PM_{2.5}$, and its components), black carbon (BC), organic carbon (OC), and combined $NO_x$/VOCs in 2013 were changed

individually to those for 2017 in each sensitivity experiment (total number of experiments = 10), and the results were compared

with those in the baseline experiment (Table S2). The second set of experiments was designed to investigate the effect of

changes in aerosols on $O_3$ concentrations via altering the photolysis rates and heterogeneous reactions (Table S3). The

individual effects of aerosol were deleted in each sensitivity experiment, and the results were compared with those in the

baseline simulation. The corresponding differences showed the effects of aerosols on $O_3$ concentrations in 2013 in terms of

photolysis rates or with respect to each heterogeneous reaction. A similar method was applied to the simulation of 2013 but

with the 2017 anthropogenic emissions, and the difference was the effect of aerosols on the $O_3$ concentrations when the

anthropogenic emissions of 2017 were applied in 2013. Finally, by comparing the results before and after the change of

emissions from 2013 to 2017, the responses of $O_3$ concentrations to changes in aerosols via altering the photolysis rates and

each heterogeneous reaction were quantified. Nineteen sensitivity experiments were performed. The third set of experiments

was designed to calculate the magnitude that the VOCs emissions in 2017 would have had to be reduced by from 2013 to

overcome the adverse effect of the changes in other pollutant emissions on $O_3$ reduction during this period. Based on the

simulation of 2013 incorporating the 2017 anthropogenic emissions of all pollutants except VOCs, the VOCs emissions were

reduced by 10%, 20%, 30% 40% and 50% in the sensitivity runs and the results were compared with those in the baseline

experiment (Table S4). By comparing the response of the 2013 $O_3$ concentration to various VOCs emissions reductions, the

required reduction of VOCs emissions was quantified.

**3 Results**

**3.1 Comparison of the simulated and observed reactive gases**

The simulated concentrations of reactive gases that are subject to significant heterogeneous reactions were compared with the

observed values for the gases $O_3$, $NO_2$, $NO_3$, $N_2O_5$, HONO, $ClNO_2$, $HO_2$, OH, and $H_2O_2$ (Table S1). Except for $O_3$ and $NO_2$,

which are measured by the regular national air monitoring network, the other gases were measured only in research-focused

field campaigns. We compiled literature-reported summer concentrations of these gases for various years and compared these

with the model-simulated values for 2013.

The uptake of $NO_2$ on wet aerosol surfaces can produce HONO in the atmosphere, which is an important source of OH radicals

via photolysis. After the update of the CMAQ model, the predicted average $NO_2$ concentration in China decreased from 19.2

ppbv to 16.6 ppbv, which came close to the observed value (15.1 ppbv). As a product of $NO_2$ uptake, the HONO concentrations

increased significantly and approached the observed values in Beijing (Wang et al., 2017a) and Guangzhou (Qin et al., 2009;

Li et al., 2012b). The decrease in $NO_2$ concentrations and increase in HONO concentrations were attributed to the increase in

heterogeneous reaction rates of $NO_2$ due to the effects of relative humidity and sunlight intensity in the updated CMAQ model

(Fu et al., 2019). Table S1 also presents the observed HONO concentrations at two coastal sites in Hong Kong (Li et al., 2018b;

Xu et al., 2015), but their concentrations were substantially underpredicted because capturing such coastal characteristics is

challenging for the model, due to its low horizontal resolution (36 km).

The simulated $NO_3$ concentration decreased slightly (~1 pptv) due to the decrease in $NO_2$ concentrations and the increase in

$\gamma_{NO_3}$ (from $10^{-4}$ to $10^{-3}$). This decrease in $NO_3$ concentrations was much smaller than the differences between the simulated

and observed values in Shanghai (Wang et al., 2013) and Guangzhou (Li et al., 2012a). Nevertheless, the simulated NO$_3$

concentration moved closer to the observed value in Shanghai (Wang et al., 2013) after the heterogeneous reactions in the

model were updated.

The parameterization of $\gamma_{N_2O_5}$ remains unchanged in the revised model. However, the decrease in NO$_2$ concentrations

described above resulted in a decrease in N$_2$O$_5$ concentrations, and thus a decrease in ClNO$_2$ concentrations. The simulated

maximum N$_2$O$_5$ concentration at the Wangdu site decreased by ~50% and thus agreed much better with the observed value

(Tham et al., 2016). The simulated maximum ClNO$_2$ concentration decreased slightly, by a margin much smaller than the

biases between the simulation and observation. Table S1 presents the observed N$_2$O$_5$ and ClNO$_2$ concentrations at a high-

altitude site on Mount Tai (Wang et al., 2017c) and a coastal site in Hong Kong (Yan et al., 2019; Tham et al., 2014). Large

differences between simulations and observations were found due to the complex terrains at these two sites, which are difficult

for our model to simulate.

The CMAQ model predicted slightly lower the concentrations of HO$_2$ and OH radicals after the incorporation of their

heterogeneous uptakes. The changes were small, probably due to the scavenging effects of aerosols being counteracted by the

increase in radical sources generated by HONO photolysis. The measured value for the concentration of HO$_2$ contains an

uncorrected contribution from RO$_2$ (Fuchs et al., 2011), which could explain in part the underestimation of HO$_2$ concentrations

that occurred when using the updated and original models. For OH radicals, the concentrations simulated by both the original

and updated models were comparable with the observed value in Wangdu (Tan et al., 2017), Beijing (Lu et al., 2013), and

Guangzhou (Lu et al., 2012). The slight decrease in the OH concentration after the update helped bring the simulation closer

to the observation.

In the original CMAQ model, the MDA8 O$_3$ concentration was overestimated by 11.4 ppbv. The bias was reduced to 6.8 ppbv

with the updated heterogeneous reactions. In addition to the greater uptake of O$_3$ on aerosol surfaces, the updated model also

includes other heterogeneous gas-aerosol reactions, weakening the atmospheric oxidation capacity and thus inhibiting O$_3$

formation.

The H$_2$O$_2$ concentration decreased substantially from ~0.8 ppbv to ~0.2 ppbv, and the simulated value agreed well with the

values recorded in Wangdu (Wang et al., 2016) and Beijing (Qin et al., 2018; Liang et al., 2013) after updating the model. Our

results suggest that the chemical transport models are likely to substantially overestimate the H$_2$O$_2$ concentration if they do not

include the sink of H$_2$O$_2$ on aerosol surfaces.

In summary, after updating the heterogeneous reactions in the CMAQ model, the simulations agreed better with the

observations, especially for concentrations of NO$_2$, HONO, O$_3$, and H$_2$O$_2$.

### 3.2 Variations in the urban and rural O$_3$ concentrations

As most of the 493 air-quality monitoring sites established in 2013 are located in urban areas (refer to Fig. S1 in Part 1 (Liu





and Wang, 2020)), the data from these stations mainly reflect the $O_3$ concentration changes in urban areas. The model simulations for the summer months from 2013 to 2017 over China give a comprehensive picture of the variations in $O_3$ concentration over the entire country. Our previous analysis based on model simulations revealed that different trends in $O_3$ concentrations existed in urban and rural areas. In this study, we quantified the trends in $O_3$ concentrations in urban and rural

areas over China using the nighttime light data from the Visible Infrared Imaging Radiometer Suite (VIIRS) Day/Night Band (DNB) (Fig. S2a). We allocated the nighttime light data to the CMAQ modeling domain and averaged the values in each modeling grid. An urban area (rural area) was regarded as a grid-point with an averaged light-value $\geq$ ($<$) 2 nanowatts cm$^{-2}$ sr$^{-1}$. Fig. S2b shows the spatial distribution of the urban and rural areas in China. The rates of changes in the MDA8 $O_3$ concentrations in urban and rural areas from 2013 to 2017 were then quantified based on the simulation results (Fig. 1). The

model predicted that the MDA8 $O_3$ concentration in urban areas increased at a rate of 0.46 ppbv per year (ppbv a$^{-1}$) (p = 0.001), while the concentration in rural areas decreased at a rate of 0.17 ppbv a$^{-1}$ (p = 0.005). Overall, the MDA8 $O_3$ concentration in China exhibited a slightly decreasing trend (0.15 ppbv a$^{-1}$, p = 0.006).

**3.3 Response of $O_3$ concentration to changes in multi-pollutant emissions**

Fig. 2 presents the spatial distribution of changes in individual pollutant emissions in 2017 relative to 2013

(http://www.meicmodel.org). Significant reductions in anthropogenic emissions of $NO_x$, CO, $SO_2$, $NH_3$, $PM_{10}$, $PM_{2.5}$, BC, and OC were found in eastern China, while the emissions in western China decreased slightly, and even increased in some areas. $NH_3$ emission, which is primarily from agriculture (Fig. S3e), generally decreased across eastern China but increased in large areas in Neimenggu and northwestern China and some scattered areas in eastern China. VOCs emissions, which have not been subject to effective control measures, increased at scattered points (mostly industrial sites) over eastern China, except for

Shandong province, where VOCs emissions decreased across the region. In summary, the emissions of $NO_x$, CO, $SO_2$, $PM_{10}$, $PM_{2.5}$, BC, and OC over mainland China were reduced by 21%, 24%, 61%, 38%, 33%, 29%, 34% in summer from 2013 to 2017, respectively (Fig. S3). In contrast, $NH_3$ emissions only decreased by 4%, and VOCs emissions increased by 5% during the same period.

Fig. 3 shows the spatial distribution of the effect of changes in these pollutant emissions on the MDA8 $O_3$ concentrations over

China between 2013 and 2017. The average changes in $O_3$ concentrations in urban and rural areas (see Fig. S2b for their locations) are shown in Fig. 4a and 4b, respectively. The decrease in $NO_x$ emissions caused an increase in $O_3$ concentrations in urban and industrial hot spots but a decrease in $O_3$ concentrations across a large swathe of rural areas (Fig. 3a). Quantitatively, the MDA8 $O_3$ concentration increased by 0.30 ppbv in urban areas and decreased by 1.08 ppbv in rural areas, due to the $NO_x$ emission reductions (Fig. 4a and b). In view of the small effects of changes in other pollutant emissions on rural $O_3$

concentrations (Fig. 4b), the decreasing trend of $O_3$ concentrations from 2013 to 2017 in rural areas was mainly ascribed to the reduction of $NO_x$ emissions, consistent with the fact that $O_3$ formation in rural areas in China is generally limited by $NO_x$



(e.g., Xing et al., 2011). The increase in $O_3$ concentrations in urban areas due to $NO_x$ reductions can be explained by two factors. First, most urban areas are in the VOCs-limited regime, where the reduction of $NO_x$ emissions reduces the NO titration effect on $O_3$, resulting in increased $O_3$ concentrations. Second, the decrease in $NO_x$ emissions can reduce the $NO_3^-$ concentration and increase the $O_3$ concentration via weakening the aerosol effects.

In the simulation of VOCs emission changes, the spatial distribution of the $O_3$ concentration closely tracked the changes in VOCs emissions (Fig. 3b). Specifically, the increase in VOCs emission caused an increase in the MDA8 $O_3$ concentrations across eastern China, except for Shandong province, where $O_3$ concentrations decreased due to the substantial reduction of VOCs emissions from the transportation sector according to the MEIC emission inventory (http://www.meicmodel.org). The simulation predicted an increase of 0.41 ppbv in the MDA8 $O_3$ concentrations from 2013 to 2017 due to the increase in VOCs emissions in urban areas (Fig. 4a). When changes in both the $NO_x$ and VOCs emissions were simulated, it was the changes in $NO_x$ emissions that primarily contributed to the changes in $O_3$ concentrations (Fig. 3c). In the simulation of changing CO emissions, the reduction of CO emissions reduced the $O_3$ concentrations across China (Fig. 3d). A particularly large decrease in $O_3$ concentrations was found in the NCP region, where both the CO emissions and their corresponding reduction were large. The CO emission reductions led to a decrease of 0.41 ppbv in the MDA8 $O_3$ concentrations in urban areas (Fig. 4a). CO is an important $O_3$ precursor and plays a similar role to VOCs in $O_3$ formation, but the changes in its emission have rarely been discussed in previous studies of the causes of variations in $O_3$ concentrations. In fact, our results indicated that the reduction of CO emissions was the only government-implemented measure that reduced $O_3$ concentrations in recent years.

In addition to the effects of $O_3$ precursors, the emissions of other pollutants can also affect $O_3$ concentrations by altering photolysis rates and the loss of reactive gases from heterogeneous reactions. The reduction of $SO_2$ emissions increased the $O_3$ concentrations across China, particularly in northern China and the Sichuan Basin (SCB) (Fig. 3e). Quantitatively, $SO_2$ emissions reductions led to an increase of 0.75 ppbv in the MDA8 $O_3$ concentrations in urban areas (Fig. 4a), which was the largest cause of $O_3$ concentration increases among all the pollutant emissions changes considered in this work. The $SO_2$ emission was reduced by more than 60% from 2013 to 2017, which resulted in a significant decrease in ambient $SO_4^{2-}$ concentrations, and increased $O_3$ concentrations by increasing the photolysis rates and retarding the loss of reactive gases from heterogeneous reactions. The reduction of $NH_3$ emissions, an important precursor of ammonium, increased the $O_3$ concentration across China in a similar way to the reduction in $SO_2$ emissions (Fig. 3f), but to a small extent, as the $NH_3$ emission was only reduced by 4%. Specifically, the increase in the MDA8 $O_3$ concentrations in urban areas due to the reduction of $NH_3$ emissions was only 0.06 ppbv (Fig. 4a), which was an insignificant fraction of the total increases in $O_3$ concentrations.

The reduction of primary PM emissions also enhanced $O_3$ formation across China, especially in the NCP and SCB regions (Fig. 3g). The MDA8 $O_3$ concentrations increased by 0.72 ppbv due to the PM emission reduction in urban areas (Fig. 4a). The effect of the changes in PM emissions on $O_3$ concentrations was comparable with that of the changes in $SO_2$ emissions, which indicated the significant $O_3$-promoting role played by reductions in both primary and secondary aerosols. BC and OC



are among the components of direct aerosol emissions, and reductions in both were found to increase the $O_3$ concentrations
(Fig. 3h and i). Although the reduction of BC emissions was smaller than the reduction in OC emissions, the increase in MDA8
$O_3$ concentrations due to the former was more significant. BC is an especially strong absorber of visible solar radiation in the
atmosphere (Ramanathan and Carmichael, 2008), and therefore greatly retards photolysis rates by reducing the solar radiation
reaching the earth's surface.

The dominant cause of $O_3$ concentration increases due to emission changes varied among regions. Fig. 4 shows the average
changes in $O_3$ concentrations due to changes in individual pollutant emissions in four megacities, Beijing, Shanghai,
Guangzhou, and Chengdu (refer to Fig. S1 in part 1 for their locations), which are the representative cities in the Beijing-
Tianjin-Hebei (BTH), Yangtze River Delta (YRD), Pearl River Delta (PRD), and SCB regions, respectively. In Beijing, $NO_x$
and PM emission reductions were the two largest causes of rising $O_3$ concentrations, followed by $SO_2$ emission reductions. Air
quality in the BTH region is a major concern and strict emission-control measures have been implemented since 2013. As a
result, the emissions of $NO_x$, $PM_{2.5}$, and $SO_2$ in BTH were reduced by 25%, 44%, and 65% from 2013 to 2017 (Fig. S4),
respectively, which were generally larger reductions than occurred in other regions (Fig. 2). In Shanghai, the increase in the
$O_3$ concentrations was mainly due to the reduction of $NO_x$ emissions and increase in VOCs emissions. This result is consistent
with the finding of Yu et al. (2019) using the Kolmogorov-Zurbenko filtering technique, who also suggested that the changes
in $O_3$ precursor emissions in the YRD contributed more to $O_3$ concentrations increases than did the decrease in $PM_{2.5}$
concentrations. In the YRD, $NO_x$ emissions decreased by 19% and that of VOCs increased by 10% from 2013 to 2017 (Fig.
S5). Meanwhile, the $PM_{2.5}$ concentration in Shanghai was relatively low in summer. As a result, the effects of the PM and $SO_2$
emission reductions were smaller than those due to the changes in $NO_x$ and VOCs emissions. In Guangzhou, the $NO_x$ emission
reduction was the dominant cause of increases in $O_3$ concentrations, while the effects of $SO_2$ and PM emission reductions on
$O_3$ concentrations were insignificant. This result can likewise be ascribed to the low concentration of $PM_{2.5}$ in summer and
relatively large reduction of $NO_x$ emissions (Fig. S6) in the PRD. In Chengdu, the PM and $SO_2$ emission reductions contributed
most to the increases in $O_3$ concentrations. The concentration of $PM_{2.5}$ in the SCB was high due to the basin topography and
high emissions of both PM and its precursors. The significant reductions of $PM_{2.5}$ (35%) and $SO_2$ (65%) emissions in the SCB
(Fig. S7) were thus the two major causes of the increase in $O_3$ concentrations there. The inter-city variations in the dominant
causes of increases in $O_3$ concentrations mean that the government should adopt additional, localized emission-reduction
measures as part of policies aimed to alleviate urban $O_3$ pollution (see section 3.5).

### 3.4 The effects of aerosol on the variations in $O_3$ concentrations

Aerosols in the atmosphere derived from direct emission and secondary formation can reduce photolysis rates and scavenge
reactive gases from heterogeneous reactions, thereby inhibiting $O_3$ formation. Fig. 5 shows the spatial distribution of changes
in the MDA8 $O_3$ concentrations due to the changes in the radiative and heterogeneous chemical effects of aerosols from 2013



to 2017 (see Methods). We isolated the effects of changes in seven heterogeneous reactions on the $O_3$ variations, and the average changes in $O_3$ concentrations in urban and rural areas are shown in Fig. 6a and 6b, respectively. As the $PM_{2.5}$ concentrations decreased substantially due to the reduction of anthropogenic pollutant emissions, the effects of aerosols on $O_3$ concentrations also decreased, which led to an increase in $O_3$ concentrations. The effects of the decrease in PM concentrations on $O_3$ concentrations were insignificant in western China. Significant increases in $O_3$ concentrations due to the decrease in

various aerosol effects were found in urban and industrial areas of eastern China, particularly the NCP and SCB regions, where pollutant emissions were high and subject to a substantial reduction in the past few years. We found that the heterogeneous chemical effect, rather than the radiative effect, contributed most to the increase in $O_3$ concentrations driven by changes in PM concentrations. Quantitatively, the changes in photolysis rates and heterogeneous reactions increased the MDA8 $O_3$ concentration by 0.30 ppbv and 2.12 ppbv in urban areas, respectively. In rural areas, the MDA8 $O_3$ concentration increased

by 0.87 ppbv via the heterogeneous chemical reactions on aerosols, while the effect of changes in photolysis rates was negligible. As for various heterogeneous reactions, the changes in individual reactions all increased the MDA8 $O_3$ concentration from 2013 to 2017. The decrease in the aerosol-sink effect of $HO_2$ contributed most to the increase in $O_3$ concentrations due to changes in PM concentrations, followed by $O_3$, $N_2O_5$, and $H_2O_2$. The effects of changes in the uptakes of $NO_2$, $NO_3$, and OH on the increase in $O_3$ concentrations were small.

The effect of the decrease in aerosol concentrations on $O_3$ concentrations varied by city. Significant effects were found in Beijing and Chengdu, where the $PM_{2.5}$ concentration was high and has subject to a large reduction by the implementation of emission-control measures. In contrast, the $PM_{2.5}$ concentration was lower in Shanghai and Guangzhou, and their $O_3$ concentrations were less affected by the decrease in aerosol concentrations.

Li et al. (2019) also investigated the effects of changes in photolysis rates and heterogeneous reactions on $O_3$ concentrations,

using the GEOS-Chem model incorporating heterogeneous reactions of nitrogen oxides and $HO_2$. They quantified the effect of changes in photolysis rates by scaling the aerosol-extinction rate using the satellite-based aerosol optical depth changes, and the effect of changes in heterogeneous reactions by scaling the aerosol surface area using the measurement-based $PM_{2.5}$ changes from 2013 to 2017. They concluded that the increase in $O_3$ concentrations due to changes in PM concentrations could be largely ascribed to the decrease in the effect of $HO_2$ heterogeneous reaction. Using a regional model and adopting different

experimental settings, our work uncovered a similar and substantial effect of $HO_2$ uptake on increases in $O_3$ concentrations due to changes in PM concentrations. In addition, with more heterogeneous reactions implemented in the CMAQ model, we found that the uptake of $O_3$ on aerosol surfaces was also important, following $HO_2$.

### 3.5 The need for concurrent reduction of anthropogenic VOCs emissions

The results in the preceding sections show that although the CO emission reductions contributed to a decrease in $O_3$

concentrations, the reductions of $SO_2$, $NO_x$, and PM emissions had a counterproductive effect on $O_3$ reductions, resulting in a





substantial increase in urban $O_3$ concentrations due to the non-linear $NO_x$-VOCs chemistry and the weakening of aerosol effects. To alleviate these negative effects of PM-targeted control policies and thereby reduce ambient $O_3$ concentrations, we found that anthropogenic VOCs emissions must also be reduced, alongside reductions in emissions of other pollutants.

Fig. 7 presents the changes in the MDA8 $O_3$ concentrations from its 2013 value, where the 2013 VOCs emissions were

decreased from 0% to 50% while the 2017 emissions of other pollutants were retained (see Methods). The MDA8 $O_3$ concentrations in the four studied megacities decrease linearly with the reduction of VOCs emissions, reflecting that $O_3$ formation in these cities is VOCs-limited. Compared with the $O_3$ concentration in 2013, the VOCs emissions would have needed to be reduced by approximately 20% to prevent increases in MDA8 $O_3$ concentrations from 2013 to 2017. This suggests that the adverse effects of the reductions of $NO_x$, $SO_2$, and PM emissions on urban $O_3$ concentrations could have been avoided

with a ~20% reduction of VOCs emissions from 2013 to 2017. The exact reductions of VOCs emissions required vary among the four megacities: Beijing (24%), Shanghai (23%), Guangzhou (20%), and Chengdu (16%). In Beijing (BTH region), the drastic reductions of $NO_x$, $SO_2$, and PM emissions would have necessitated a more substantial reduction of VOCs emissions to counteract the increases in concentrations of $O_3$. In Shanghai (YRD region) and Guangzhou (PRD region), the increase in $O_3$ concentrations due to the reductions in $NO_x$ emissions also calls for a significant reduction in VOCs emissions. In Chengdu

(SCB region), a smaller VOCs emission reduction is needed because of the relatively small increase in $O_3$ concentrations due to changes in other emissions. We also found that the required percentage reductions of VOCs emissions in each city were comparable with the actual percentage reductions in $NO_x$ emissions (25%, 19%, 18%, and 14% for Beijing, Shanghai, Guangzhou, and Chengdu, respectively), suggesting that similar percentage reductions of VOCs and $NO_x$ would have prevented the increase in $O_3$ concentrations from 2013 to 2017.

Our results have important implications for air-pollution control policy in the coming years. In 2018, the Chinese government issued a Three-Year Action Plan (2018–2020) mandating further reductions of national $SO_2$ and $NO_x$ emissions by at least 15% by the year 2020 compared with those in the year 2015, and an 18% reduction in ambient $PM_{2.5}$ concentrations in cities currently not compliant with China's $PM_{2.5}$ standards (http://www.gov.cn/zhengce/content/2018-07/03/content_5303158.htm). This implies that if VOCs emissions are not reduced in the near future, the $O_3$ pollution in major cities will continue to worsen.

Therefore, VOCs emission controls should be implemented together with the PM-targeted measures.

**4 Conclusions**

This study has quantified the effects of changes in individual pollutant emissions from anthropogenic activities on the summer surface $O_3$ concentrations over China from 2013 to 2017. The reduction in $NO_x$ emissions was found to increase the urban $O_3$ concentrations but reduce the rural $O_3$ concentrations, with little effect on the overall $O_3$ burden in China. Past control measures,

while successful in reducing the concentrations of primary pollutants and particulate matter, have worsened the urban $O_3$

problem due to the non-linear chemistry of O₃ and the complex effects of aerosols. We demonstrate that comparable percentage

reductions in anthropogenic VOCs to that achieved for NOₓ could have prevented the increases in O₃ concentrations. We thus

conclude that VOCs controls should be implemented in current and future emission-reduction measures to improve the overall

air quality. In view of the importance and complexity of the uptake of reactive gases on aerosol surfaces, more research should

be conducted in this area.

**Author contributions**

T.W. initiated the research. Y.M.L. and T.W. designed the research framework. Y.M.L. modified the model and performed

model simulations. T.W. and Y.M.L. analyzed the results and wrote the paper.

**Competing interests**

The authors declare that they have no conflict of interest.

**Code/Data availability**

The code or data used in this study are available upon request from Yiming Liu (yming.liu@polyu.edu.hk) and Tao Wang

(cetwang@polyu.edu.hk).

**Acknowledgments**

This work was supported by the Hong Kong Research Grants Council (T24-504/17-N) and the National Natural Science

Foundation of China (91844301). We would like to thank Prof. Qiang Zhang from Tsinghua University for providing the

emission inventory, and Dr. Xiao Fu from The Hong Kong Polytechnic University for sharing the model codes of HONO

sources.

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



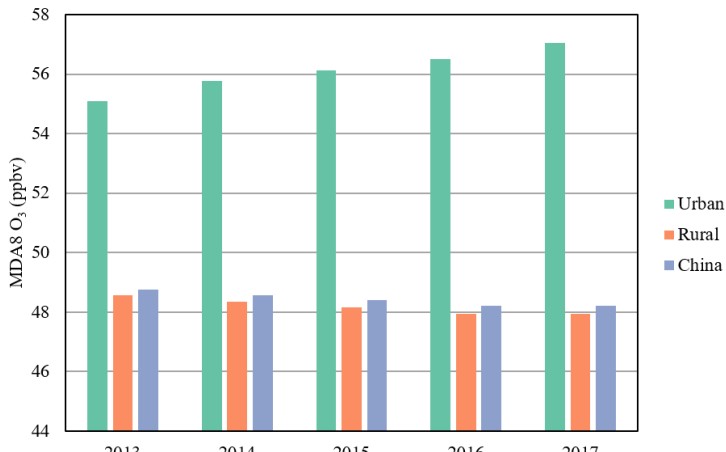

**Figure 1: Trends of simulated MDA8 O₃ concentrations averaged in urban and rural areas and all of China in summer (June-August) from 2013 to 2017. See Fig. S2b for the locations of urban and rural areas over China.**


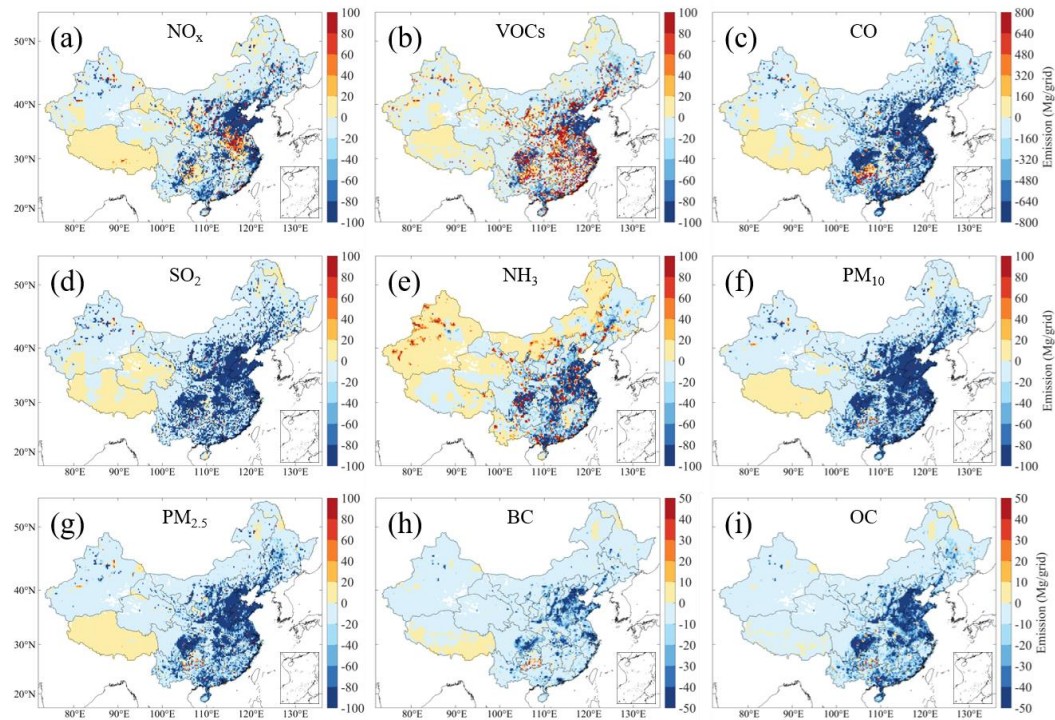


**Figure 2: Spatial distributions of changes in anthropogenic pollutant emissions in the summer of 2017 relative to that of 2013, including (a) NOₓ, (b) VOCs, (c) CO, (d) SO₂, (e) NH₃, (f) PM₁₀, (g) PM₂.₅, (h) BC, and (i) OC. Emission data are obtained from Multi-resolution Emission Inventory for China (MEIC; http://www.meicmodel.org).**






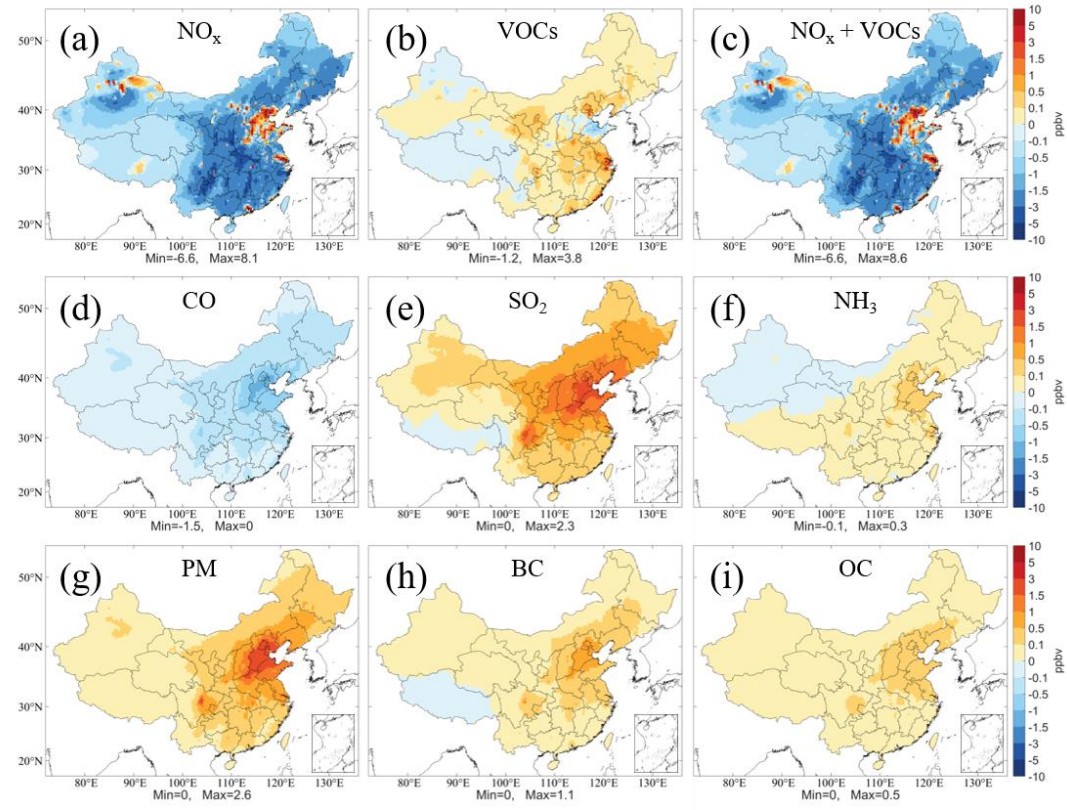

**Figure 3: Spatial distribution of the simulated MDA8 O₃ concentrations responding to the changes of individual pollutant emissions in summer from 2013 to 2017, including (a) NOₓ, (b) VOCs, (c) NOₓ and VOCs, (d) CO, (e) SO₂, (f) NH₃, (g) PM, (h) BC, and (i) OC.**





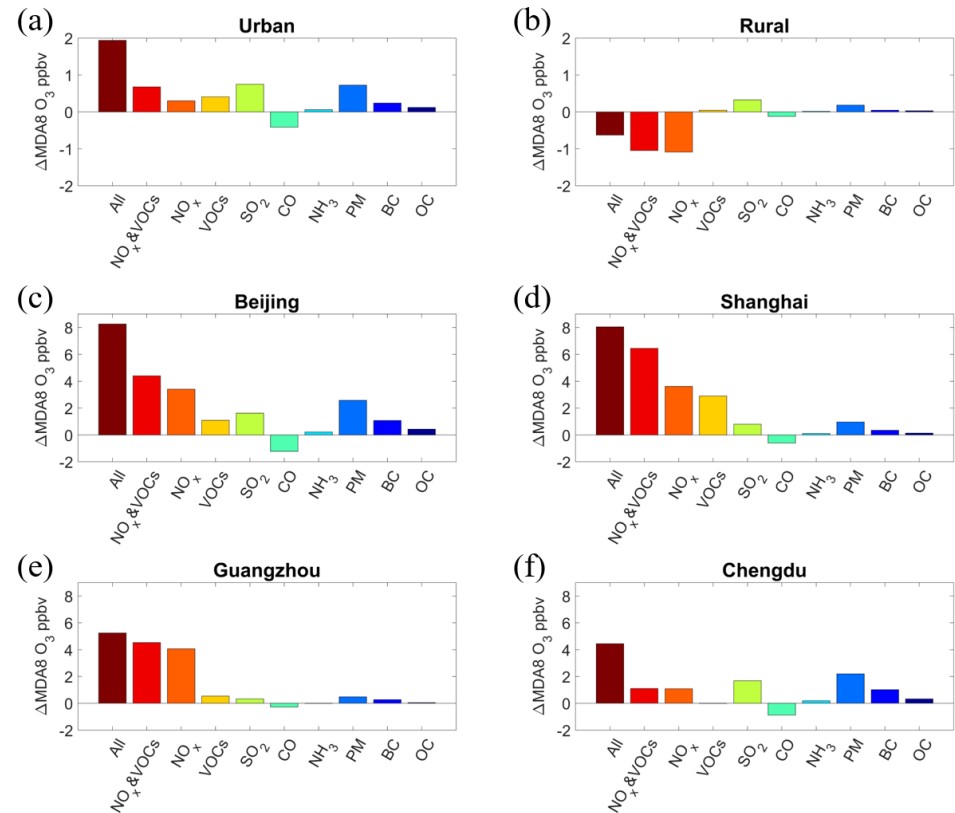

**Figure 4: Response of the simulated MDA8 O₃ concentrations to the changes in individual pollutant emissions in summer from 2013 to 2017 in (a) the urban areas, (b) the rural areas, (c) Beijing, (d) Shanghai, (e) Guangzhou, and (f) Chengdu. See Fig. S1 in part 1 (Liu and Wang, 2020) for the locations of the four megacities.**



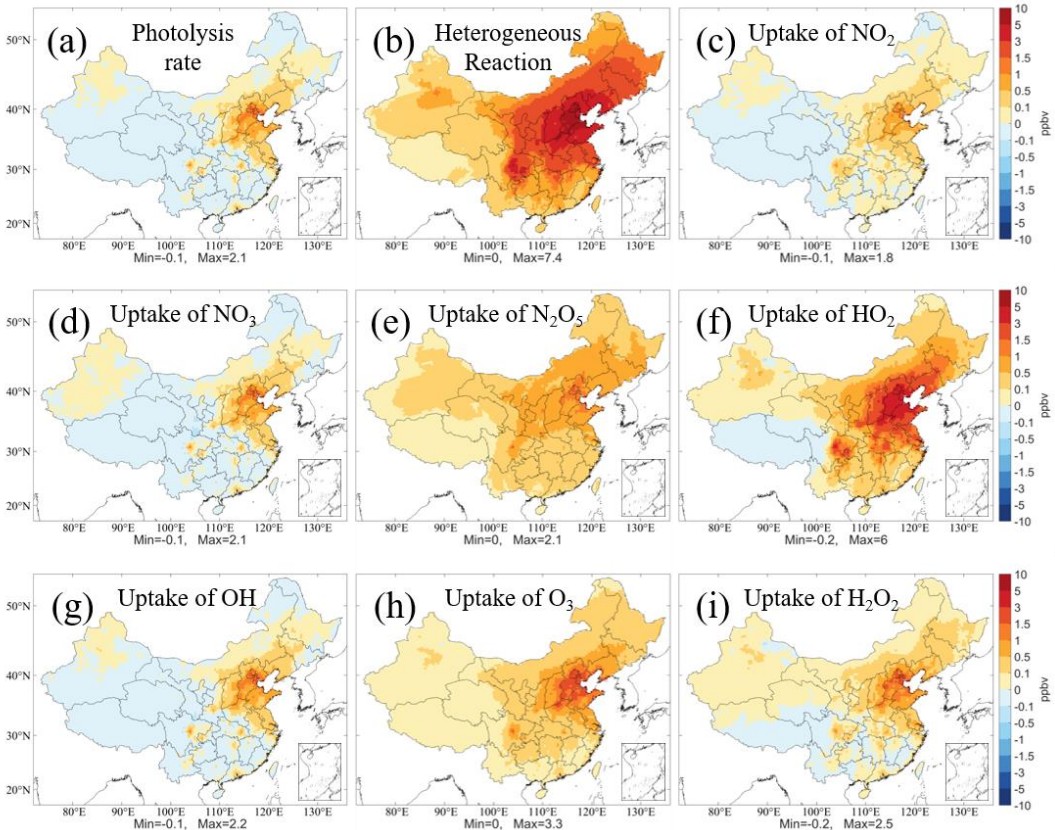

**Figure 5: Spatial distribution of the simulated MDA8 O₃ concentrations responding to the changes in the effects of aerosol in summer from 2013 to 2017 (see Methods). The aerosol affects the O₃ concentrations via altering the (a) photolysis rates, (b) all heterogeneous reactions, and individual heterogeneous reactions, namely the uptake of (c) NO₂, (d) NO₃, (e) N₂O₅, (f) HO₂, (g) OH, (h) O₃, and (i) H₂O₂.**




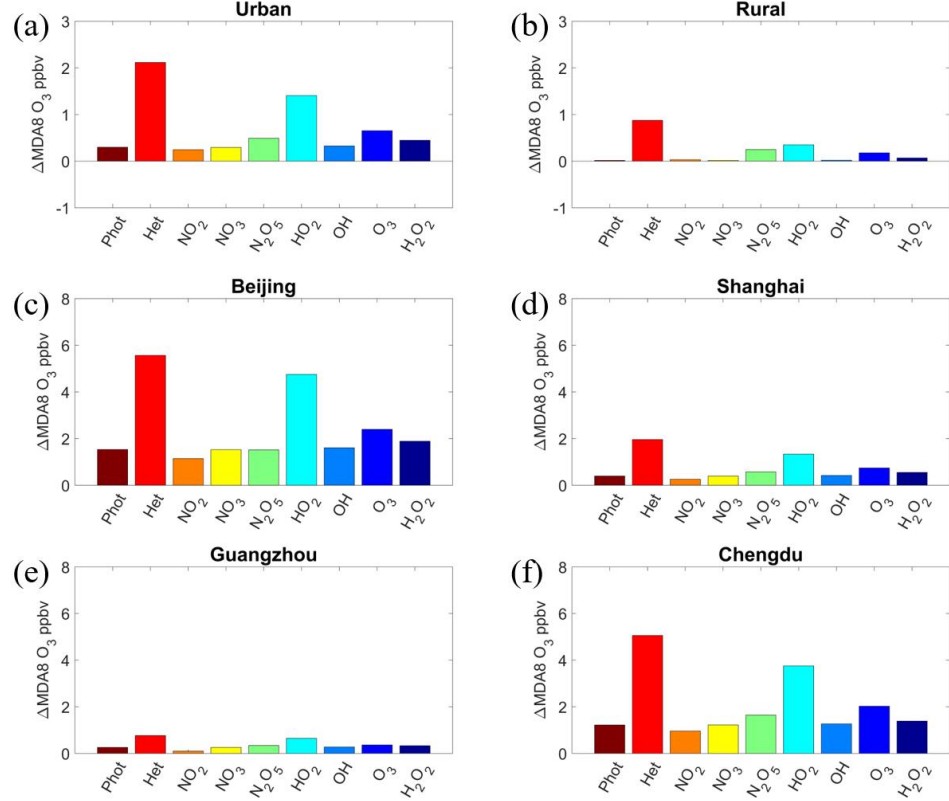

**Figure 6: Response of the simulated MDA8 O₃ concentrations to the changes in the effects of aerosols in summer from 2013 to 2017 in (a) the urban area, (b) the rural area, (c) Beijing, (d) Shanghai, (e) Guangzhou, and (f) Chengdu. The aerosol affects the O₃ concentration via altering the photolysis rates (Phot), all heterogeneous reactions (Het), and individual heterogeneous reactions, namely the uptake of NO₂, NO₃, N₂O₅, HO₂, OH, O₃, and H₂O₂.**







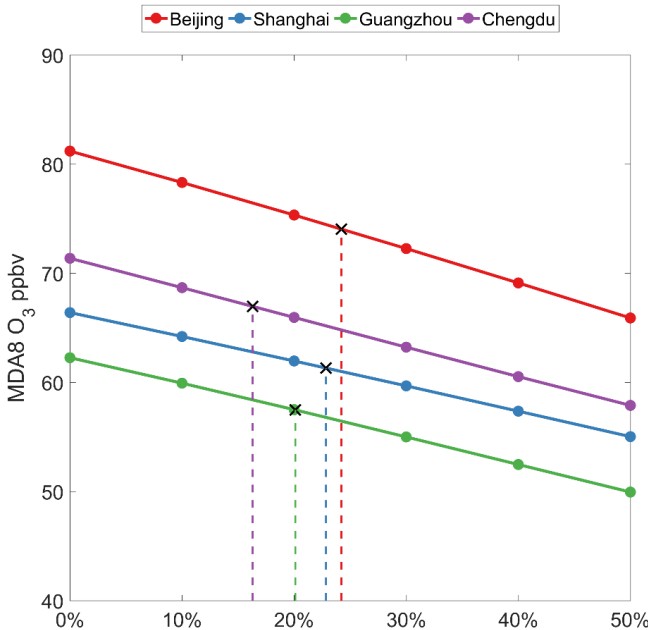

**Figure 7: Response of simulated MDA8 O$_3$ concentrations with 2017 emissions (except for VOCs) to the reductions of anthropogenic VOCs from the 2013 level in summer in Beijing, Shanghai, Guangzhou, and Chengdu. The black crosses depict the MDA8 O$_3$ concentrations in 2013 and the required reduction of VOCs emissions in 2017 to maintain the 2013 O$_3$ concentration in each city.**