# Peer review of "Worsening urban ozone pollution in China from 2013 to 2017 – Part 2: The effects of emission changes and implications for multi-pollutant control"

_Atmospheric Chemistry and Physics, 2020_

## Referee Comment (RC1) · Anonymous Referee #1 · 10 Mar 2020

This is a very well written paper, which provides a thorough analysis of the impacts of gas and PM emission controls on ozone formation across China. The paper is appropriate for ACP and it works well with the companion paper that is also under review with ACPD. As described below, there are a few items that need to be addressed, after which the paper would be suitable for publication in ACP.

Major Comments:

1) Lines 230-232 Here some context needs to be provided for these trends, and some

evaluation against observations is warranted. Figure S1 shows that observed ozone increased by 18% across all urban areas with ozone monitors. However, the model indicates that urban ozone increased from 55 to 57 ppbv, which is just a 3.6 % increase, a rate that is five times less than the observed rate. Why are the modeled trends so low compared to the observed trends, and which processes are being missed by the models? To help the readier understand the discrepancy between the model and observations the authors need to directly compare the model to observations. For example they can compare the modeled trend in the grid cell (or cells) above Beijing to all of the monitors with data from 2013-2017. They can make similar plots for the other urban areas of YRD, PRD and SCB

Across all of China the model predicts a very small ozone decrease of 0.6 ppbv, or just 1%. It's difficult to believe that this tiny decrease has any real meaning. How is the p-value (0.006) so low? What kind of statistical test was used? To have such a tiny decrease with such a low p-value indicates that the signal-to-noise ratio is very high, which implies that there is very little interannual variability. But Part I of this study shows that meteorology creates substantial interannual variability.

2) This science paper strays into the realm of policy recommendations, as follows: Line 308-310 "The inter-city variations in the dominant causes of increases in O3 concentrations mean that the government should adopt additional, localized emission-reduction measures as part of policies aimed to alleviate urban O3 pollution (see section 3.5)."

Line 343 "3.5 The need for concurrent reduction of anthropogenic VOCs emissions"

Line 370 "Therefore, VOCs emission controls should be implemented together with the PM-targeted measures."

"Line 377-379 We thus conclude that VOCs controls should be implemented in current and future emission-reduction measures to improve the overall air quality."

I understand that the authors want their paper to be beneficial for improving air quality

in China, and their results will certainly be useful. However, the recommendations will have to be re-phrased so that this science paper does not sound like a policy document. Fortunately, this is a straightforward editorial process. Instead of saying what the government "should" do, the authors can say something like: "Recent emission controls across China have not reduced ozone and have actually increased ozone in urban areas. If the government wishes to adopt new emissions control policies that will reduce ozone in urban and rural areas we propose the following recommendations for VOC controls…." By phrasing it like this, your paper offers very useful options to the government without sounding like a policy paper.

3) This study focuses on summer, but did the authors also look at ozone changes during the winter months? TOAR-Climate (Gaudel et al., 2018) compares surface ozone trends at non-urban sites across North America, during 2000-2014, a period of deceasing NOx emissions. Ozone decreases across much of the continent in summer, but increases in winter (see their Figures 13, 14 and 15). I wonder if a similar pattern has occurred across China in winter.

Gaudel, A., et al. (2018), Tropospheric Ozone Assessment Report: Present-day distribution and trends of tropospheric ozone relevant to climate and global atmospheric chemistry model evaluation, Elem Sci Anth, 6(1):39, DOI: https://doi.org/10.1525/elementa.291

Minor Comments

Line 286-288 Here the authors state that, in general, BC has a major impact on photolysis rates. But the overall conclusion from this study is that the impact of PM reductions on ozone production is mainly through the changes in heterogeneous chemistry, with the impact on photolysis rates being secondary. Given the conclusions of the study it would be a good idea to provide some additional context for the impact of BC on photolysis rates and ozone production.

Line 104 Here and elsewhere, there is no such word as "uptakes". To make it plural

you can use "uptake rates"

Line 143 This sentence would sound better as: "The companion paper (Part 1; (Liu and Wang, 2020)) presented validation results. . ."

Line 208 Change "observation" to "observations"

Line 209 Here and throughout the paper, when mentioning a trace gas value in units of ppbv, then the quantity must be referred to as a mixing ratio, and not a concentration, which has units of mass per volume.

Line 331 has should be was ". . .where the PM2.5 concentration was high and WAS subject. . ."
* * *

---

## Referee Comment (RC2) · Anonymous Referee #2 · 17 Mar 2020

The authors examined the effects of 2013-2017 changes in anthropogenic emissions on summertime ozone pollution over China, and they found that the emission controls for reducing aerosols have worsened urban ozone through the non-linear chemistry of ozone and the complex effects of aerosols. The current increasing trend of ozone in China is of great concern and this topic is well within the scope of ACP journal. The authors here present a very comprehensive study, and the manuscript is well structured. The estimated effects of emissions of individual chemical species on ozone are valuable for air quality planning in China. I would recommend it to be accepted after

addressing the following comments.

-Heterogeneous uptake of ozone. The simulated increases in ozone from this pathway (Fig.5h) are high over regions with high PM2.5 concentrations other than regions with high levels of mineral dust. I am wondering if you are applying this effect for all the aerosols or just on dust particles. The uptake of ozone by aerosols are only well documented for mineral aerosol. Bauer et al. (2004) also suggested that the lower limit of uptake coefficient (3x10-6) seems to be more appropriate for global modeling.

-The updated model will decrease NO2 concentration, and it compares better with surface NO2 in summer 2013. But the model also has low biases for summers 2014-2017 (in Table 2 of the companion paper). Please have more explanations on this.

-The simulated decreases in NO3 and N2O5 (Lines 185-195) could be also induced by decreased ozone in the updated simulation.

The Conclusion section needs to be rewritten. Currently the 9 lines of conclusion are not a good summary of what have been done in the manuscript. Quantitative conclusions should be given in both Abstract and Conclusion section.

The manuscript is not clear about the impact of boundary conditions of chemical species on simulated O3 in China. Ideally the chemical boundary conditions are different for 2013 and 2017, considering the differences in anthropogenic emissions and in meteorology outside the model domain. How would these differences at the boundary influence simulated changes in O3 in China over 2013-2017? Some discussions can be added in Conclusion section.

---

## Author Comment (AC1) · 14 Apr 2020

This is a very well written paper, which provides a thorough analysis of the impacts of gas and PM emission controls on ozone formation across China. The paper is appropriate for ACP and it works well with the companion paper that is also under review with ACPD. As described below, there are a few items that need to be addressed, after which the paper would be suitable for publication in ACP.

Response: We thank the referee for providing a thoughtful and detailed review of our paper. The referee's comments have helped to improve this manuscript. Below, we provide a point-by-point response to the referee's comments and summarize the changes that have been made in the revised manuscript.

**Major Comments:**

[Comment]: 1. Lines 230-232 Here some context needs to be provided for these trends, and some evaluation against observations is warranted. Figure S1 shows that observed ozone increased by 18% across all urban areas with ozone monitors. However, the model indicates that urban ozone increased from 55 to 57 ppbv, which is just a 3.6 % increase, a rate that is five times less than the observed rate. Why are the modeled trends so low compared to the observed trends, and which processes are being missed by the models? To help the readier understand the discrepancy between the model and observations the authors need to directly compare the model to observations. For example they can compare the modeled trend in the grid cell (or cells) above Beijing to all of the monitors with data from 2013-2017. They can make similar plots for the other urban areas of YRD, PRD and SCB

Across all of China the model predicts a very small ozone decrease of 0.6 ppbv, or just 1%. It's difficult to believe that this tiny decrease has any real meaning. How is the p-value (0.006) so low? What kind of statistical test was used? To have such a tiny decrease with such a low p-value indicates that the signal-to-noise ratio is very high, which implies that there is very little interannual variability. But Part I of this study shows that meteorology creates substantial interannual variability.

Response: Thanks for this valuable comment. The simulated MDA8 O3 increase (~2 ppbv) in the nightlight-classified urban areas from 2013 to 2017 is much lower than the average increase observed at 493 sites in 74 cities (~9 ppbv). The discrepancy can be explained as follows. The urban areas determined using the nightlight data are not exactly the same as those 493 sites and cover some rural areas (with decreasing ozone) and additional small townships (see Fig R1 below). If we match the model output with the observation sites, then the model can capture 57% of the average rate of increase at those sites, as shown in Figure R2 below. We had compared the simulated and observed MDA8 O3 changes in Beijing (BTH), Shanghai (YRD), Guangzhou (PRD), and Chengdu (SCB) in Section 3.3 of Part 1 (Liu and Wang, 2020). The result showed that the model could also generally capture the changes in observed MDA8 O3 in different cities. We have added some texts in the revised manuscript to clarify this discrepancy.

The p value was calculated using the F-test statistical method. As shown in Fig. 1, the MDA8 O3 mixing ratio across all of China did present a small decreasing trend with a high confidence level (p=0.006). Part 1 of this study showed the large variability of meteorological impacts on O3 in regions and years, but this regional variability can be 'averaged out' over the whole China, leading to a clearer ozone trend. The very small ozone decreases in China indicated that the ozone concentration has leveled off in recent years, attributable to the decrease in large rural areas due to the NOx emission reduction. The recently published studies also supported our model predicted ozone decreases in rural areas of eastern China. Wang et al. (2019) revealed no significant change in O3 levels observed at a coastal site (Hok Tsui) in South China in the outflow of air mass from eastern China during 2007-2018. Xu et al. (2020) reported decreasing O3 mixing ratios from 2013 to 2016 at two rural sites in BTH (Shangduanzi) and YRD (Linan).

Revision in the main text:

**1) Line 227-242:**

"The model predicted that the MDA8 O3 mixing ratio in urban areas increased at a rate of 0.46 ppbv per year (ppbv a-1) (p = 0.001). This simulated increase (~2 ppbv from 2013 to 2017) in the nightlight-classified urban areas is much lower than the average increase observed at 493 sites in 74 cities (~9 ppbv, Fig. S1d). The discrepancy can be explained as follows. The urban areas determined using the nightlight data are not exactly the same as those 493 sites and cover some rural areas (with decreasing ozone) and additional small townships (Fig. S3). When we matched the modeled locations to the 493 observation sites, the model captured 57% of the rate of increase of MDA8 O3 averaged at those sites (see Fig. S3 in Part 1 (Liu and Wang, 2020)). Part 1 also showed a large variability of meteorological impacts on O3 in different regions (e.g., Beijing, Shanghai, Guangzhou, and Chengdu), and the simulated overall urban O3 trend with a high confidence level (p = 0.001) suggests that this regional variability in meteorological impact can be 'averaged out', leading to a clearer urban O3 trend driven by emission changes.

The simulated MDA8  $O_3$  mixing ratio in rural areas decreased at a rate of 0.17 ppbv a-1 (p = 0.005), which is supported by the recently reported rural ozone trends in China. Wang et al. (2019c) revealed no significant change in  $O_3$  levels observed at a coastal site (Hok Tsui) in South China in the outflow of air mass from eastern China during 2007-2018. More recently, Xu et al. (2020) reported decreasing  $O_3$  mixing ratios from 2013 to 2016 at two rural sites in BTH (Shangduanzi) and YRD (Linan). Overall, MDA8  $O_3$  mixing ratio in China exhibited a slightly decreasing trend (0.15 ppbv a-1, p = 0.006) due to the decrease in a large rural area, which suggested that the ozone concentration has leveled off in recent years."

**Reference:**

- Liu, Y., and Wang, T.: Worsening urban ozone pollution in China from 2013 to 2017 Part 1: The complex and varying roles of meteorology, Atmos. Chem. Phys. Discuss., 2020, 1-28, 10.5194/acp-2019-1120, 2020.
- Wang, T., Dai, J., Lam, K. S., Nan Poon, C., and Brasseur, G. P.: Twenty-Five Years of Lower Tropospheric Ozone Observations in Tropical East Asia: The Influence of Emissions and Weather Patterns, 46, 11463-11470, 10.1029/2019g1084459, 2019c.
- Xu, X., Lin, W., Xu, W., Jin, J., Wang, Y., Zhang, G., Zhang, X., Ma, Z., Dong, Y., Ma, Q., Yu, D., Li, Z., Wang, D., and Zhao, H.: Long-term changes of regional ozone in China: implications for human health and ecosystem impacts, Elem Sci Anth, 8, 13, 10.1525/elementa.409, 2020.

Figure R1 (Figure S3) Spatial distribution of the urban and rural areas in land areas of China identified by using the nighttime light data. The yellow cross "+" represents the locations of 493 environmental monitoring stations in 74 cities since 2013. BTH, YRD, PRD, SCB are the Beijing-Tianjin-Hebei, Yangtze River Delta, Pearl River Delta, and Sichuan Basin regions, respectively.

Figure R2 (Fig. S3 in Part 1 (Liu and Wang, 2020)) Changes in observed and simulated summer surface MDA8 O3 mixing ratios averaged in 493 sites of 74 cities during 2013-2017 relative to those of 2013.

[Comment]: 2. This science paper strays into the realm of policy recommendations, as follows: Line 308-310 "The inter-city variations in the dominant causes of increases in  $O_3$  concentrations mean that the government should adopt additional, localized emission-reduction measures as part of policies aimed to alleviate urban  $O_3$  pollution (see section 3.5)."

Line 343 "3.5 The need for concurrent reduction of anthropogenic VOCs emissions"

Line 370 "Therefore, VOCs emission controls should be implemented together with the PM-targeted measures."

"Line 377-379 We thus conclude that VOCs controls should be implemented in current and future emission-reduction measures to improve the overall air quality."

I understand that the authors want their paper to be beneficial for improving air quality in China, and their results will certainly be useful. However, the recommendations will have to be re-phrased so that this science paper does not sound like a policy document. Fortunately, this is a straightforward editorial process. Instead of saying what the government "should" do, the authors can say something like: "Recent emission controls across China have not reduced ozone and have actually increased ozone in urban areas. If the government wishes to adopt new emissions control policies that will reduce ozone in urban and rural areas we propose the following recommendations for VOC controls. . .." By phrasing it like this, your paper offers very useful options to the government without sounding like a policy paper.

Response: It was our intention to emphasize the policy implications of the results. We understand the referee's viewpoint. In the revised region, we have rephrased these descriptions and made it not reading like a policy paper. Revision in the main text:

1) Line 316-318:

"The inter-city variations in the dominant causes of increases in  $O_3$  concentrations suggest that if the government wishes to alleviate urban  $O_3$  pollution, they can adopt additional, localized emission-reduction measures as part of policies (see section 3.5)."

- 2) Line 350:
  - "3.5 The anthropogenic VOCs emission control to reduce O3"
- 3) Line 376-377:

"Therefore, we suggest VOCs emission controls be implemented together with the PM-targeted measures in order to alleviate the urban O3 pollution."

4) Line 391-392:

"We thus recommend that VOCs control be implemented in current and future emission-reduction measures to improve the overall air quality."

[Comment]: 3. This study focuses on summer, but did the authors also look at ozone changes during the winter months? TOAR-Climate (Gaudel et al., 2018) compares surface ozone trends at non-urban sites across North America, during 2000-2014, a period of deceasing NOx emissions. Ozone decreases across much of the continent in summer, but increases in winter (see their Figures 13, 14 and 15). I wonder if a similar pattern has occurred across China in winter.

Gaudel, A., et al. (2018), Tropospheric Ozone Assessment Report: Present- day distribution and trends of tropospheric ozone relevant to climate and global atmospheric chemistry model evaluation, Elem Sci Anth, 6(1):39, DOI: https://doi.org/10.1525/elementa.291

Response: We had also examined the ozone trend in winter. Figure R3 below depicts the observed ozone changes during the winter months (January, February, and December) from 2013-2017 at the same 493 cities. Like summer, the averaged MDA8 O3 concentration also presented an overall increasing trend in winter. As the present study

focuses on the photochemically active summer season, we do not discuss the winter result.

---

## Author Comment (AC2) · 14 Apr 2020

The authors examined the effects of 2013-2017 changes in anthropogenic emissions on summertime ozone pollution over China, and they found that the emission controls for reducing aerosols have worsened urban ozone through the non-linear chemistry of ozone and the complex effects of aerosols. The current increasing trend of ozone in China is of great concern and this topic is well within the scope of ACP journal. The authors here present a very comprehensive study, and the manuscript is well structured. The estimated effects of emissions of individual chemical species on ozone are valuable for air quality planning in China. I would recommend it to be accepted after addressing the following comments.

Response: We thank the referee for providing a thoughtful review of our paper and the recognition of our work. The referee's comments have helped to improve this manuscript. Below, we provide a point-by-point response to the referee's comments and summarize the changes that have been made in the revised manuscript.

[Comment]: 1. Heterogeneous uptake of ozone. The simulated increases in ozone from this pathway (Fig.5h) are high over regions with high $PM_{2.5}$ concentrations other than regions with high levels of mineral dust. I am wondering if you are applying this effect for all the aerosols or just on dust particles. The uptake of ozone by aerosols are only well documented for mineral aerosol. Bauer et al. (2004) also suggested that the lower limit of uptake coefficient (3x10-6) seems to be more appropriate for global modeling.

Response: We applied an uptake coefficient of $O_3$ ($\gamma_{O_3}$) of $1 \times 10^{-5}$ to all aerosols in our model simulations. Previous laboratory and measurement studies indicated that $\gamma_{O_3}$ varied in a wide range on different aerosols: $10^{-6}$-$10^{-4}$ on dust (Michel et al., 2002, 2003; Hanisch and Crowley, 2003), up to $10^{-4}$ on sodium chloride aerosols (Abbatt and Waschewsky, 1998), and $10^{-5}$-$10^{-3}$ on soot particles (Longfellow et al., 2000). Most previous modeling studies adopted $1 \times 10^{-5}$ (Liao et al., 2004; Liao and Seinfeld, 2005; Pozzoli et al., 2008), while one recommended a lower value ($3 \times 10^{-6}$) for dust particles (Bauer et al., 2004). We think our choice of $10^{-5}$ is reasonable.

Revision in the main text:

1) Line 134-138:

"Previous laboratory and measurement studies of the heterogeneous reaction of $O_3$ have given a wide range of ($\gamma_{O_3}$, from $10^{-6}$ to $10^{-4}$ on dust (Michel et al., 2002, 2003; Hanisch and Crowley, 2003), up to $10^{-4}$ on sodium chloride aerosol (Abbatt and Waschewsky, 1998), from $10^{-5}$ to $10^{-3}$ on soot particles (Longfellow et al., 2000). Most previous modeling studies adopted $1 \times 10^{-5}$ (Liao et al., 2004; Liao and Seinfeld, 2005; Pozzoli et al., 2008), while one recommended a lower value ($3 \times 10^{-6}$) for dust particles (Bauer et al., 2004). We applied $10^{-5}$ to the uptake of $O_3$ on all the aerosols in our simulation."

Reference:

Abbatt, J. P. D., and Waschewsky, G. C. G.: Heterogeneous Interactions of HOBr, $HNO_3$, $O_3$, and $NO_2$ with Deliquescent NaCl Aerosols at Room Temperature, The Journal of Physical Chemistry A, 102, 3719-3725, 10.1021/jp980932d, 1998.

Bauer, S. E., Balkanski, Y., Schulz, M., Hauglustaine, D. A., and Dentener, F.: Global modeling of heterogeneous chemistry on mineral aerosol surfaces: Influence on tropospheric ozone chemistry and comparison to observations, J Geophys Res-Atmos, 109, 10.1029/2003jd003868, 2004.

Hanisch, F., and Crowley, J. N.: Ozone decomposition on Saharan dust: an experimental investigation, Atmos. Chem. Phys., 3, 119-130, 10.5194/acp-3-119-2003, 2003.

Liao, H., Seinfeld, J. H., Adams, P. J., and Mickley, L. J.: Global radiative forcing of coupled tropospheric ozone and

aerosols in a unified general circulation model, J Geophys Res-Atmos, 109, 2004.

Liao, H., and Seinfeld, J. H.: Global impacts of gas-phase chemistry-aerosol interactions on direct radiative forcing by anthropogenic aerosols and ozone, 110, 10.1029/2005jd005907, 2005.

Longfellow, C. A., Ravishankara, A. R., and Hanson, D. R.: Reactive and nonreactive uptake on hydrocarbon soot: $HNO_3$, $O_3$, and $N_2O_5$, 105, 24345-24350, 10.1029/2000jd900297, 2000.

Michel, A. E., Usher, C. R., and Grassian, V. H.: Heterogeneous and catalytic uptake of ozone on mineral oxides and dusts: A Knudsen cell investigation, 29, 10-11-10-14, 10.1029/2002gl014896, 2002.

Michel, A. E., Usher, C. R., and Grassian, V. H.: Reactive uptake of ozone on mineral oxides and mineral dusts, Atmos. Environ., 37, 3201-3211, https://doi.org/10.1016/S1352-2310(03)00319-4, 2003.

Pozzoli, L., Bey, I., Rast, S., Schultz, M. G., Stier, P., and Feichter, J.: Trace gas and aerosol interactions in the fully coupled model of aerosol-chemistry-climate ECHAM5-HAMMOZ: 1. Model description and insights from the spring 2001 TRACE-P experiment, 113, 10.1029/2007jd009007, 2008.

[Comment]: 2. The updated model will decrease $NO_2$ concentration, and it compares better with surface $NO_2$ in summer 2013. But the model also has low biases for summers 2014- 2017 (in Table 2 of the companion paper). Please have more explanations on this.

Response: In Part 1 (Liu and Wang, 2020), we had given one main reason for the low biases of simulated $NO_2$ for summers 2014-2017, which is related to the fact that the $NO_2$ concentrations in the national observation network were measured using the catalytic conversion method, which overestimates $NO_2$, especially during periods of active photochemistry (Xu et al., 2013; Zhang et al., 2017; Fu et al., 2019). Therefore, the updated model has improved simulations for all the years.

Reference:

Fu, X., Wang, T., Zhang, L., Li, Q., Wang, Z., Xia, M., Yun, H., Wang, W., Yu, C., Yue, D., Zhou, Y., Zheng, J., and Han, R.: The significant contribution of HONO to secondary pollutants during a severe winter pollution event in southern China, Atmos. Chem. Phys., 19, 1-14, 10.5194/acp-19-1-2019, 2019.

Liu, Y., and Wang, T.: Worsening urban ozone pollution in China from 2013 to 2017 – Part 1: The complex and varying roles of meteorology, Atmos. Chem. Phys. Discuss., 2020, 1-28, 10.5194/acp-2019-1120, 2020.

Xu, Z., Wang, T., Xue, L. K., Louie, P. K. K., Luk, C. W. Y., Gao, J., Wang, S. L., Chai, F. H., and Wang, W. X.: Evaluating the uncertainties of thermal catalytic conversion in measuring atmospheric nitrogen dioxide at four differently polluted sites in China, Atmos. Environ., 76, 221-226, 10.1016/j.atmosenv.2012.09.043, 2013.

Zhang, L., Li, Q. Y., Wang, T., Ahmadov, R., Zhang, Q., Li, M., and Lv, M. Y.: Combined impacts of nitrous acid and nitryl chloride on lower-tropospheric ozone: new module development in WRF-Chem and application to China, Atmos Chem Phys, 17, 9733-9750, 10.5194/acp-17-9733-2017, 2017.

[Comment]: 3. The simulated decreases in $NO_3$ and $N_2O_5$ (Lines 185-195) could be also induced by decreased ozone in the updated simulation.

Response: Thanks for this comment which we agree. We have modified the statements in the revised manuscript.

Revision in the main text:

1) Line 187:

   "The simulated $NO_3$ mixing ratio decreased slightly (~1 pptv) due to the decrease in $NO_2$ and $O_3$ mixing ratios"

2) Line 192-193:

"the decrease in $NO_2$ and $O_3$ levels resulted in a decrease in $N_2O_5$"

[Comment]: 4. The Conclusion section needs to be rewritten. Currently the 9 lines of conclusion are not a good summary of what have been done in the manuscript. Quantitative conclusions should be given in both Abstract and Conclusion section.

Response: We didn't intend to repeat the content of the abstract. Nonetheless, we understand the referee's view and have added more content in the revised conclusion section.

Revision in the main text:

1) Line 379-393:

"This study has quantified the effects of changes in pollutant emissions from anthropogenic activities on the summer surface $O_3$ concentrations over China from 2013 to 2017. The control measures, while successful in reducing the concentrations of primary pollutants and particulate matter, were found to increase urban $O_3$ but reduce rural $O_3$; overall, the $NO_x$ emission reduction has helped to contain total ozone production in China. The reduction in $NO_x$ emission and slight increase in VOC emissions led to ozone increase in urban areas due to the non-linear chemistry of $O_3$, and the large reductions in PM and $SO_2$ emissions contributed to urban ozone increase resulting from the complex effects of aerosols on radiation and chemical reactions. Among the primary PM components, the emission decrease in BC increased $O_3$ more than that for OC despite its smaller reduction compared to OC, resulting from BC being a strong absorber of solar radiation. The dominant causes of the urban ozone increase due to emission change varied among different cities, and they were $NO_x$ and PM in Beijing, $NO_x$ and VOC in Shanghai, $NO_x$ in Guangzhou, and PM and $SO_2$ in Chengdu. For the aerosol effects, the decrease in heterogeneous uptake of reactive gases was more important than the increase in photolysis rates. Only the CO emission cut helped to decrease urban ozone. Our results show that comparable percentage reductions in anthropogenic VOCs to that achieved for $NO_x$ could have prevented the increases in urban $O_3$ concentrations. We thus recommend that VOCs control be implemented in current and future emission-reduction measures to improve the overall air quality. In view of the importance and complexity of the uptake of reactive gases on aerosol surfaces, more research should be conducted in this area."

[Comment]: 5. The manuscript is not clear about the impact of boundary conditions of chemical species on simulated $O_3$ in China. Ideally the chemical boundary conditions are different for 2013 and 2017, considering the differences in anthropogenic emissions and in meteorology outside the model domain. How would these differences at the boundary influence simulated changes in $O_3$ in China over 2013-2017? Some discussions can be added in Conclusion section.

Response: The impact of boundary conditions has been simulated and discussed in Part 1 (Liu and Wang, 2020), so we will not repeat them in the present paper. Briefly, the chemical boundary conditions for the CMAQ model were derived from the results of the MOZART global model, and they varied in 2013-2017. We found that the impact of the long-range transport from outside the CMAQ modeling domain to China contributed to the increase in MDA8 $O_3$ in China during 2013-2017, especially on the Qinghai-Tibetan Plateau (with an increase of 1 to 4 ppbv). More discussions are presented in Part 1, Section 3.6 (Liu and Wang, 2020).

Revision in the main text:

1) Line 78-80:

"The role of meteorological variation and total emission changes, the effect of changes in individual meteorological factors, and the impact of changes in long-range transport of $O_3$ and its precursors from outside

the modeling domain, are discussed in a companion paper, Part 1 (Liu and Wang, 2020)."

Reference:

Liu, Y., and Wang, T.: Worsening urban ozone pollution in China from 2013 to 2017 – Part 1: The complex and varying roles of meteorology, Atmos. Chem. Phys. Discuss., 2020, 1-28, 10.5194/acp-2019-1120, 2020.